# Modulating Chitinase in the QS Biosensor Strain CV026: Do Not Forget to Release Carbon Catabolite Repression. Comment on Deryabin et al. Quorum Sensing in *Chromobacterium subtsugae* ATCC 31532 (Formerly *Chromobacterium violaceum* ATCC 31532): Transcriptomic and Genomic Analyses. *Microorganisms* 2025, *13*, 1021

**DOI:** 10.3390/microorganisms13102235

**Published:** 2025-09-24

**Authors:** Alex Leite Pereira, Fernanda Favero, Angelo Henrique Lira Machado

**Affiliations:** 1Faculdade de Ciências e Tecnologias em Saúde, Campus de Ceilândia, Universidade de Brasília, Brasília 72220-275, DF, Brazil; 2Programa de Pós-Graduação em Medicina Tropical, Faculdade de Medicina, Universidade de Brasília, Brasília 70910-900, DF, Brazil; 3Instituto de Química, Universidade de Brasília, Campus Universitário Darcy Ribeiro, Brasília 70910-900, DF, Brazil

**Keywords:** chitinolytic activity, quorum sensing, carbon catabolite repression

## Abstract

Chitinolytic activity is a well-documented phenotype controlled by quorum sensing (QS) in *Chromobacterium* strains but also regulated by carbon catabolite repression mechanisms. This work comprehensively reviews scientific literature on chitinolytic activity, reinforcing the need to use a minimal culture medium supplemented with chitin for assays testing chitinolytic activity modulated by QS in *Chromobacterium* strains.

## 1. Introduction

We read with great interest the recent article by Deryabin et al. addressing transcriptional and genomic analyses in the quorum sensing (QS) strain ATCC 31532 [1]. However, we were surprised to find that the authors stated in their concluding remarks that the results call into question “the chitinase assay in QS modulation experiments using the ATCC 31532-derived CV026 biosensor strain” published by our group [2].

Unfortunately, we suspect some experimental issues, including phenotypic modulation in bacterial culture and cloning approaches applied in *Chromobacterium* QS research, may have been misunderstood by Deryabin et al. [1].

## 2. Discussion

In the seminal series of papers published by Stewart and Williams’ group in the mid-1990s [3,4], the mini-Tn5 double mutant strain [∆-cviI, ∆-vioS] CV026, derived from *C. subtsugae* ATCC 31532, was phenotypically demonstrated to be defective in N-(hexanoyl)-L-homoserine lactone (C6-HSL) production. Consequently, it was shown to be completely deficient in both violacein production and chitinolytic activity [3,4,5]. Furthermore, supplementation of CV026 cultures with culture supernatants from the *C. subtsugae* wild-type strain or 10 μM of synthetic C6-HSL restored violacein and chitinase production to levels comparable to those of the parental wild-type strain (Figure 1) [3,4,5].

Since the initial experiments published on QS in *Chromobacterium* strains, violacein production is routinely achieved by growing bacterial strains in nutrient-rich medium such as Luria broth (LB) (tryptone 10 g/L, yeast extract 5 g/L and NaCl 10 g/L). Growth of wild-type strains of *C. violaceum* ATCC 12472 or *C. subtsugae* ATCC 31532 in LB under aeration or on LB agar produces detectable violacein, as does the QS-mutant strain CV026 when supplemented with exogenous C6-HSL [2,3,4,5,6,7,8].

However, a completely different culture environment must be established to enable the expression of chitinolytic activity in experimental models exploiting *Chromobacterium* spp. strains.

The extracellular degradation of chitin and subsequent uptake of chitin-derived oligosaccharides is an energetically demanding process for bacterial cells [9,10]. Therefore, prioritizing the consumption of simple sugars before expending energy to degrade complex polysaccharides, such as chitin, is energetically advantageous for bacteria, and allows for faster growth. Mechanisms that downregulate costly metabolic pathways in response to preferred carbon sources are known as carbon catabolite repression (CCR) [9,10].

CCR mechanisms have been shown to control chitinolytic activity in bacterial systems, including Gammaproteobacteria and Firmicutes species. In *Pseudomonas* spp., GacA (global regulator A) is a high-level regulator expressed during restricted bacterial growth under limited nutrient availability, activating regulatory RNAs (*rmsY* and *rmsZ*) to modulate carbon assimilation [11]. Studies have demonstrated that GacA controls chitinase activity in *P. fluorescens* and *P. aeruginosa* [12,13]. Additionally, QS regulation of chitinase has also been demonstrated in *P. aeruginosa* PAO1 [14]. However, evidence supports that *Pseudomonas* QS systems are under the modulation of the GacA system [15].

In *Serratia marcescens*, CCR is primarily modulated by the cAMP and its receptor protein (CRP), which triggers the expression of catabolite-repressed genes. Carbohydrate transport is achieved by the phosphoenolpyruvate (PEP)-dependent carbohydrate phosphotransferase system (PTS) [16]. The major CCR signal from PTS is the phosphorylation state of sugar-specific permeases (enzymes IIA-D). When preferred carbon sources are limiting, phosphorylated EIIA activates adenylate cyclase to generate cAMP, which in turn activates CRP and thereby releases CCR [17]. Studies have shown that mutations in CRP, EI or EIIA impair the expression of chitinase and chitin-binding protein in *S. marcescens* [17,18,19].

*Vibrio cholerae* maintains control over chitinase genes through molecular circuits similar to those in *S. marcescens* [10]. Furthermore, in *V. cholerae*, chitin oligosaccharides serve as both a nutrient source and an environmental signal that induces a strong transcriptional response [10].

The expression of chitinolytic activity in *Bacillus thuringiensis* is well characterized and demonstrates inhibition by glucose and induction in the presence of chitin. Chitinase genes are negatively regulated by CcpA (catabolic control protein A), which binds to and blocks *cre* (catabolic response element) sequences in target promoters when glucose is available [20].

Control over chitinase activity in *Chromobacterium* spp. has been characterized in less molecular detail. However, it is well documented that chitinase expression is directly controlled by QS-modulated transcriptional activator (CviR) (in *C. violaceum* strain 12472), induced in the presence of chitin (in *C. subtsugae* ATCC 31532), and restored in QS-mutant strain CV026 upon supplementation with C6-HSL [4,21].

CCR control over chitinolytic activity in *C. subtsugae* ATCC 31532 and its derivative mutant strain CV026 has also been demonstrated by Chernin et al. [4]. Using phenotypic assays on agar plates to visualize chitinolytic activity and extracellular protein extracts to measure enzymatic activity, the authors demonstrated that *C. subtsugae* ATCC 31532 produces chitinase only when grown in minimal media (LB diluted to 10% [vol/vol]) containing chitin (0.2% colloidal chitin), as does strain CV026 when supplemented with exogenous C6-HSL. Furthermore, neither strain showed constitutive chitinolytic activity when chitin was replaced by glucose or sucrose in the growth medium, regardless of HSL supplementation [4].

Given the role of CCR in regulating chitinase activity, short-incubation culture assays (up to 24–36 h) designed to detect chitinase activity in *Chromobacterium* strains require the growth of bacterial cells in minimal medium supplemented with colloidal chitin (0.2 to 1% *w*/*v*). This approach for detecting chitinase activity in *Chromobacterium* spp. models (ATCC 31532, ATCC 12472 or CV026) has been systematically reproduced in studies published by independent groups [4,5,22,23] and was reproduced in our previous article [2].

Despite its necessity for chitinase expression, a minimal culture medium supplemented with chitin was not utilized by Deryabin et al. [1]. As a result, their transcriptomic approach excluded chitinase genes from the set of QS-upregulated genes in *C. subtsugae* ATCC 31532 [1]. This result led the authors to inaccurately state that QS modulation of chitinase does not occur in *C. subtsugae* ATCC 31532, unlike *C. violaceum* ATCC 12472 [1]. Moreover, these inaccurate findings have served as the basis for their claim challenging the validity of QS experiments involving the ATCC 31532-derived strain CV026, which our group had previously published [2]

To leave no doubt in this discussion, QS modulation of chitinolytic activity in *C. subtsugae* ATCC 31532 has been demonstrated by genetic experiments [5]. In 2017, Devescovi et al. revealed that the repressor protein VioS adds another regulatory layer to the QS system [5]. The authors reported the negative regulation exerted by VioS on QS-mediated upregulation of the *vioA* promoter as well as on the chitinolytic activity of *C. subtsugae*. As part of their study, the authors also developed a *cviR* mutant strain (31532CVIR, *cviR*::Gm from *C. subtsugae* ATCC 31532) in which chitinase activity was abolished [5]. Additionally, it was demonstrated that VioS functions as a repressor of violacein production in *C. violaceum* ATCC 12472 when expressed in *trans*, confirming that QS modulation in these two model strains is functionally quite similar (except for cognate autoinducers: C6-HSL in strain ATCC 31532 and C10-HSL in strain ATCC 12472) [5].

Even though modulation of chitinase activity by both CCR and QS effectors in ATCC 31532 and CV026 strains is well-established through different phenotypic assays (Table 1) [2,4,5], direct transcriptomic approaches under conditions that ensure the release of CCR remain to be fully explored.

## 3. Conclusions

In our opinion, the validity of our experimental demonstration of QS modulation of chitinase activity in CV026 remains robust [2]. The results are valid because they align with extensive scientific research on CV026 model as a biosensor for QS.

## Figures and Tables

**Figure 1 microorganisms-13-02235-f001:**
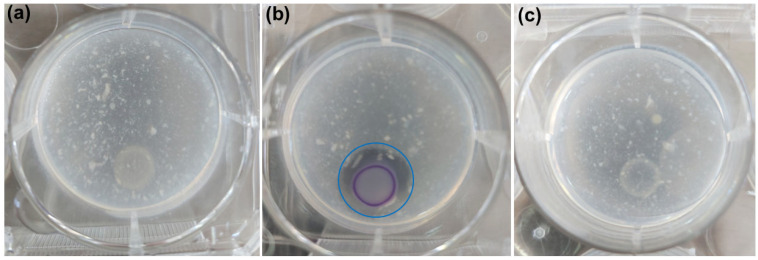
Quorum sensing-mediated induction of chitinase activity in CV026. Colonies were grown for 48 h at 28 °C on minimal medium agar (10% [vol/vol] LB) supplemented with 0.2% colloidal chitin. (**a**) Negative control without exogenous autoinducer C6-HSL. CV026 did not exhibit either chitin hydrolysis or violacein production. (**b**) Positive control with addition of 10 µM C6-HSL (QS induction). CV026 developed a distinct chitin hydrolysis halo and produced violacein. The chitin hydrolysis zone is indicated by a blue-outlined circle. (**c**) Inhibition assay with 10 µM C6-HSL and 32 µM chlorolactone (QS inhibitor). CV026 did not exhibit chitin hydrolysis or violacein production.

**Table 1 microorganisms-13-02235-t001:** Overview of experimental approaches and findings on QS-regulated chitinase in *Chromobacterium* ssp. and derivative strains.

Findings	Experimental Approach	Tested Strain	Ref.
**1.** Chitinase activity was induced by QS autoinducer C6-HSL, chitinase activity was blocked in presence of glucose or sucrose, and the expression of chitinase enzymes was induced by C6-HSL	Endogenous expression of chitinase in minimal medium supplemented with chitin, chitinase control under QS autoinducers, chitinase control under carbon catabolite repression, and enzyme detection after protein resolution on SDS-PAGE	ATCC 31532 CV026	[4]
**2.** Chitinase activity was abolished in *cviR* knockout strain, and chitinase down regulation by QS repressor protein VioS	Endogenous expression of chitinase in minimal medium supplemented with chitin, knockout and complementation strains	ATCC 31532 *cviR*::Gm knockout strain *vioS*::Km knockout strain	[5]
**3.** Chitinase gene (*CV_4240*) was activated by QS regulator CviR	Heterologous expression of *cviR* and *P_CV_4240_*::*luxCDABE* reporter system in *E. coli*	ATCC 12472	[21]
**4.** Chitinase activity induced by QS autoinducer C6-HSL, C6-HSL-induced chitinase blocked by QS antagonist chlorolactone	Endogenous expression of chitinase in minimal medium supplemented with chitin, and QS modulation by autoinducers and antagonists	CV026	[2]

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
