# Peer review of "(untitled)"

_microorganisms, 2025, doi:10.3390/microorganisms13102235_

Round 1
Reviewer 1 Report
Comments and Suggestions for Authors Pereira et al. successfully leverage extensive literature evidence and sound logic to deliver a powerful rebuttal of Deryabin et al.'s critique, effectively highlighting the critical methodological pitfall (ignoring CCR, using unsuitable media) in the latter's study. The core message regarding the necessity of appropriate culture conditions is clearly articulated and substantiated.
This is a high-quality academic commentary. The authors, Pereira et al., have successfully utilized a substantial foundation of literature and rigorous logical reasoning to effectively refute the challenges posed by Deryabin et al. to their previous work. They have also highlighted a critical methodological flaw in the opposing study (the failure to consider CCR and the use of inappropriate culture medium). The core argument of the article – emphasizing the paramount importance of correct experimental conditions – is presented clearly and powerfully, offering significant reference value for researchers in the relevant field. The article is professional, rigorous, and successfully defends the reliability of their prior research conclusions. I believe this commentary meets publication standards and is acceptable.
My specific opinion:
This commentary provides a valid argument regarding carbon catabolite repression (CCR) in chitinase studies but requires major revision due to significant limitations. The rebuttal narrowly focuses on CCR while failing to address Deryabin et al.'s broader genomic criticisms of CV026 or alternative technical explanations for their negative results. It over-relies on foundational studies (Chernin 1998, Devescovi 2017) that themselves lack quantitative validation of chitinase regulation and overlooks critical context like CV026's documented genomic instability. The reproduced Figure 1 remains qualitatively presented without scale bars or quantitative comparison to Deryabin's conditions, weakening its evidentiary value. Furthermore, the extensive review of CCR mechanisms across bacteria (~70% of text) dilutes the Chromobacterium-specific rebuttal. To strengthen its contribution, the authors must: 1) tighten focus to directly counter Deryabin's methodology with quantitative evidence, 2) acknowledge limitations in key supporting literature, 3) contextualize strain-specific reproducibility concerns, and 4) resolve ambiguous terminology regarding culture conditions.
Author Response
Reviewer 1: Pereira et al. successfully leverage extensive literature evidence and sound logic to deliver a powerful rebuttal of Deryabin et al.'s critique, effectively highlighting the critical methodological pitfall (ignoring CCR, using unsuitable media) in the latter's study. The core message regarding the necessity of appropriate culture conditions is clearly articulated and substantiated.
This is a high-quality academic commentary. The authors, Pereira et al., have successfully utilized a substantial foundation of literature and rigorous logical reasoning to effectively refute the challenges posed by Deryabin et al. to their previous work. They have also highlighted a critical methodological flaw in the opposing study (the failure to consider CCR and the use of an inappropriate culture medium). The core argument of the article – emphasizing the paramount importance of correct experimental conditions – is presented clearly and powerfully, offering significant reference value for researchers in the relevant field. The article is professional, rigorous, and successfully defends the reliability of their prior research conclusions. I believe this commentary meets publication standards and is acceptable.
My specific opinion:
This commentary provides a valid argument regarding carbon catabolite repression (CCR) in chitinase studies, but requires major revision due to significant limitations. The rebuttal narrowly focuses on CCR while failing to address Deryabin et al.'s broader genomic criticisms of CV026 or alternative technical explanations for their negative results. It over-relies on foundational studies (Chernin 1998, Devescovi 2017) that themselves lack quantitative validation of chitinase regulation and overlooks critical context like CV026's documented genomic instability. The reproduced Figure 1 remains qualitatively presented without scale bars or quantitative comparison to Deryabin's conditions, weakening its evidentiary value. Furthermore, the extensive review of CCR mechanisms across bacteria (~70% of text) dilutes the Chromobacterium-specific rebuttal. To strengthen its contribution, the authors must:
1) tighten focus to directly counter Deryabin's methodology with quantitative evidence,
Response to Reviewer 1
Dear Reviewer,
The recognition of CCR as the main controller of catabolic phenotypes in bacteria is considered by those who study bacterial physiology to be a paradigm supported by a wide collection of scientific evidence.
Any study of endogenous expression addressing the consumption of non-preferred carbon sources must first consider, in favour of scientific coherence, the catabolic repression exerted by CCR.
We submit this academic commentary to report to the scientific community on the unjustified criticisms of Deryabin et al., which are based on a lack of methodological and scientific coherence: that is, the employment of a minimum culture medium to induce CCR release and chitin supplementation are essential requirements for testing chitinolytic activity in Chromobacterium species.
Once this physiological condition is set, our phenotypic assay unequivocally and visually demonstrates the QS control over chitinase activity in CV026, testing not only the QS-inducing molecule C6-HSL, but also the QS antagonist chlorolactone.
Specifically regarding chitinase activity in Chromobacterium ssp., we insist that we provided in our response a considerable body of previously published evidence that supports our opinion and endorses the methodological coherence of our experiment, which was criticized on insufficient grounds and without due consideration of the existing literature.
In order to clarify this point, we have created Table 1 listing molecular and phenotypic evidence that supports our position. This presentation of experimental evidence is intended to allow the reader to assess the reproducibility of QS-mediated chitinase expression across ATCC strains 12472 and 31532 and its derivative strains.
Table 1 - Overview of experimental approaches and findings on QS-regulated chitinase in Chromobacterium ssp. and derivative strains.
|
Findings |
Experimental approach |
Tested strain |
Ref. |
|
Chitinase activity was induced by QS autoinducer C6-HSL
Chitinase activity was blocked in presence of glucose or sucrose
Expression of chitinase enzymes was induced by C6-HSL |
Endogenous expression of chitinase in minimal medium supplemented with chitin
Chitinase control under QS autoinducers
Chitinase control under carbon catabolite repression
Enzyme detection after protein resolution on SDS-PAGE |
ATCC 31532
CV026 |
[4] |
|
Chitinase activity was abolished in cviR knockout strain
Chitinase down regulation by QS repressor protein VioS |
Endogenous expression of chitinase in minimal medium supplemented with chitin
Knockout and complementation strains |
ATCC 31532
cviR::Gm knockout strain
vioS::Km knockout strain |
[5] |
|
Chitinase gene (CV_4240) was activated by QS regulator CviR |
Heterologous expression of cviR and PCV_4240::luxCDABE reporter system in E. coli
|
ATCC 12472 |
[21] |
|
Chitinase activity was induced by QS autoinducer C6-HSL
C6-HSL-induced chitinase was blocked by QS antagonist chlorolactone |
Endogenous expression of chitinase in minimal medium supplemented with chitin
QS modulation by autoinducers and antagonists |
CV026 |
[2] |
Furthermore, a careful re-evaluation of the data from Stauff et al. (DOI: 10.1128/jb.05125-11) is necessary to avoid drawing misleading conclusions about potential differences in the QS-controlled gene expression profiles between strains ATCC 31532 and ATCC 12472, as previously suggested by Deryabin et al. Stauff et al.’s findings were based on the heterologous expression of cviR and QS-promoter::luxCDABE fusion reporter system (PCV_4240::luxCDABE), both cloned in an E. coli strain. In this system, Stauff et al. simply identified genes under the control of QS-responsive promoters, including chitinase genes, without the confounding effect of endogenous CCR. By contrast, Deryabin et al. combined these data with their own findings on endogenous gene expression repressed by CCR to conclude that the endogenous expression of chitinase is markedly different in these two Chromobacterium species.
Now, we realised that the review of CCR could indeed be shorter. We agree and have reduced discussions on this topic.
2) acknowledge limitations in key supporting literature,
We have followed your suggestion, and the period below was included in the manuscript.
“Even though modulation of chitinase activity by both CCR and QS effectors in ATCC 31532 and CV026 strains is well-established through different phenotypic assays [2,4,5], direct transcriptomic approaches under conditions that ensure the release of CCR remain to be fully explored.”
3) contextualize strain-specific reproducibility concerns, and
Table 1 has been added to the manuscript to allow the reader to assess the reproducibility of QS-mediated chitinase endogenous expression across strains ATCC 12472 and 31532, as well as their derivative strains, whenever a minimal medium is used to bypass CCR.
4) resolve ambiguous terminology regarding culture conditions.
We have revised the ambiguous terminology concerning culture conditions.
We are grateful for your valuable revision and welcome this opportunity for a frank and high-level academic discussion. It is our firm belief that our work is dedicated to advancing the understanding of scientific findings, while acknowledging inherent limitations.
Reviewer 2 Report
Comments and Suggestions for Authors
I have gone through the manuscript. Basically it is a comment made by Pereira et al. in response to Deryabin et al. To match and compare the work and see the significance and severity of this comment, I have to read all the three papers including Pereira et al. own prior study of Favero et al.
After reading all three, I felt that authors have presented a scientifically sound argument that QS regulates chitinase production in Chromobacterium strains, and that carbon catabolite repression can mask this regulation. In my opinion it is valid and well supported through literatures.
However, on the other hand after reading the Deryabin et al. paper and comparing the two approaches, in my opinion it is very clear why Deryabin et al’s conclusions differ from Favero et al’s results. Deryabin et al. looked at gene expression in a presumably nutrient-rich medium without chitin. Consequently and obviously they did not see any difference and assumed QS had no effect on chitinase in that strain. This I believe was a methodological oversight. They did not test the bacterium under conditions where it could express chitinase. However, Favero et al., on the other hand, provided the necessary context for chitinase expression. This is basically the problem with the Deryabin’s experimental design, and not a flaw with Favero et al’s approach.
Therefore, Pereira et al. are justified in proving the Deryabin’s claim wrong, as the criticisms by Deryabin et al. are not well-justified
One minor suggestion regarding the terminology. The authors refer to ATCC 31532 sometimes as C. subtsugae and sometimes as C. violaceum ATCC 31532. This could confuse readers who are unaware that those are the same strain. It will be better and more consistent to stick with one name (C. subtsugae ATCC 31532, formerly C. violaceum) throughout.
Though the authors narrative and point are scientifically sound what they are claiming, but the manuscript has a number of grammatical and typographical issues that needs thorough proofreading. For example:
Chromobacterium spelling is wrong in one place “Cromobacterium” and in another place “Chromobaterium”.
Line 51: “Since the initials experiments……….” It should be “initial”
Line 53: LB medium composition???? It looks like authors have missed the NaCl. Please correct the composition.
“in our cited article [2]” should be “in our previous article [2]”
There is no need of the abbreviations list at the end, as this is not at all relevant to this comment. What is the point of putting MDPI, DOAJ??? Instead, if you want, you can use QS, CCR, AHL, etc……
I think authors can reduce the content on the background of CCR……….
What I also feel is that one major point is misleading in the comment. In the first few line in introduction, you have mentioned “addressing transcriptional and genomic analyses in the quorum sensing (QS) biosensor strain CV026 [1]”……however, Deryabin’s et al study was more about the wild type 31532 and not CV026 itself. Therefore, you must revise the sentence.
Author Response
Reviewer 2: I have gone through the manuscript. Basically it is a comment made by Pereira et al. in response to Deryabin et al. To match and compare the work and see the significance and severity of this comment, I have to read all the three papers including Pereira et al. own prior study of Favero et al.
After reading all three, I felt that authors have presented a scientifically sound argument that QS regulates chitinase production in Chromobacterium strains, and that carbon catabolite repression can mask this regulation. In my opinion it is valid and well supported through literatures.
However, on the other hand after reading the Deryabin et al. paper and comparing the two approaches, in my opinion it is very clear why Deryabin et al’s conclusions differ from Favero et al’s results. Deryabin et al. looked at gene expression in a presumably nutrient-rich medium without chitin. Consequently and obviously they did not see any difference and assumed QS had no effect on chitinase in that strain. This I believe was a methodological oversight. They did not test the bacterium under conditions where it could express chitinase. However, Favero et al., on the other hand, provided the necessary context for chitinase expression. This is basically the problem with the Deryabin’s experimental design, and not a flaw with Favero et al’s approach.
Therefore, Pereira et al. are justified in proving the Deryabin’s claim wrong, as the criticisms by Deryabin et al. are not well-justified
One minor suggestion regarding the terminology. The authors refer to ATCC 31532 sometimes as C. subtsugae and sometimes as C. violaceum ATCC 31532. This could confuse readers who are unaware that those are the same strain. It will be better and more consistent to stick with one name (C. subtsugae ATCC 31532, formerly C. violaceum) throughout.
Though the authors narrative and point are scientifically sound what they are claiming, but the manuscript has a number of grammatical and typographical issues that needs thorough proofreading. For example:
Chromobacterium spelling is wrong in one place “Cromobacterium” and in another place “Chromobaterium”.
Line 51: “Since the initials experiments……….” It should be “initial”
Line 53: LB medium composition???? It looks like authors have missed the NaCl. Please correct the composition.
“in our cited article [2]” should be “in our previous article [2]”
There is no need of the abbreviations list at the end, as this is not at all relevant to this comment. What is the point of putting MDPI, DOAJ??? Instead, if you want, you can use QS, CCR, AHL, etc……
I think authors can reduce the content on the background of CCR……….
What I also feel is that one major point is misleading in the comment. In the first few line in introduction, you have mentioned “addressing transcriptional and genomic analyses in the quorum sensing (QS) biosensor strain CV026 [1]”……however, Deryabin’s et al study was more about the wild type 31532 and not CV026 itself. Therefore, you must revise the sentence.
Response to Reviewer 2
Dear Reviewer,
Thank you for your valuable review and for providing us with the opportunity to have this important scientific discussion. We have corrected the typographical errors you pointed out, as well as the incorrect citation of the CV026 strain at the beginning of the article and the abbreviations. We have also reduced the content on the background of CCR.
We appreciate your helpful comments.
Reviewer 3 Report
Comments and Suggestions for Authors
Nicely written manuscript on chitinase expression illustrating a very important point that a minimal culture medium supplemented with chitin is essential for testing for chitinolytic activity. The authors have briefly reviewed (in their Discussion) relevant supporting material. Nice figure and list of relevant references. No changes in the manuscript are suggested.
Author Response
Reviewer 3: Nicely written manuscript on chitinase expression illustrating a very important point that a minimal culture medium supplemented with chitin is essential for testing for chitinolytic activity. The authors have briefly reviewed (in their Discussion) relevant supporting material. Nice figure and list of relevant references. No changes in the manuscript are suggested.
Response to Reviewer 3
Dear Reviewer,
We are grateful for your review of our manuscript. We are pleased to know that our paper has been well-evaluated by our peers. Thank you.
Reviewer 4 Report
Comments and Suggestions for Authors
The comment by Pereira et al. presents a comprehensive literature review to address and clarify the partial conclusions drawn by Deryabin et al., emphasizing that chitinolytic activity in Chromobacterium is regulated by both quorum sensing and carbon catabolite repression. Although this manuscript provides some literature evidence, I recommend publication of this comment manuscript following major revisions.
- It would be beneficial to include new molecular and transcriptomic data (e.g. gene expression analysis) to support the conclusion besides citation of the previously published literature.
- The authors refer to plate-based chitinase and violacein phenotypic assays multiple times, which are valid though but lack molecular specificity, therefore do not directly address transcriptomic differences reported by Deryabin et al.
- Please check spelling consistency (e.g. Cromobacterium spp. in line 57).
Author Response
Reviewer 4: The comment by Pereira et al. presents a comprehensive literature review to address and clarify the partial conclusions drawn by Deryabin et al., emphasizing that chitinolytic activity in Chromobacterium is regulated by both quorum sensing and carbon catabolite repression. Although this manuscript provides some literature evidence, I recommend publication of this comment manuscript following major revisions.
It would be beneficial to include new molecular and transcriptomic data (e.g. gene expression analysis) to support the conclusion besides citation of the previously published literature.
The authors refer to plate-based chitinase and violacein phenotypic assays multiple times, which are valid though but lack molecular specificity, therefore do not directly address transcriptomic differences reported by Deryabin et al.
Please check spelling consistency (e.g. Cromobacterium spp. in line 57).
Response to Reviewer 4
Dear Reviewer,
We are grateful for your valuable revision and welcome this opportunity for a frank and high-level academic discussion.
The recognition of CCR as the main controller of catabolic phenotypes in bacteria is considered by those who study bacterial physiology to be a paradigm supported by a wide collection of scientific evidence. Any study of endogenous expression addressing the consumption of non-preferred carbon sources must first consider, in favor of scientific coherence, the catabolic repression exerted by CCR.
We submit this academic commentary to report to the scientific community on the unjustified criticisms of Deryabin et al., which are based on a lack of methodological and scientific coherence: that is, the employment of minimum culture medium to induce CCR release and chitin supplementation are essential requirements for testing chitinolytic activity in Chromobacterium species.
Once this physiological condition is set, our phenotypic assay unequivocally and visually demonstrates the QS control over chitinase activity in CV026, testing not only the QS-inducing molecule C6-HSL, but also the QS antagonist chlorolactone.
We acknowledge the limitations of phenotypic assays and we have added to manuscript the statement below:
“Even though modulation of chitinase activity by both CCR and QS effectors in ATCC 31532 and CV026 strains is well-established through different phenotypic assays (table 1) [2,4,5], direct transcriptomic approaches under conditions that ensure the release of CCR remain to be fully explored.”
In order to clarify this point, we have created table 1 listing molecular and phenotypic evidence that supports our position. This presentation of experimental evidence is intended to allow the reader to assess the reproducibility of QS-mediated chitinase expression across ATCC strains 12472 and 31532 and its derivative strains, whenever a minimal medium is used to bypass CCR.
Table 1 - Overview of experimental approaches and findings on QS-regulated chitinase in Chromobacterium ssp. and derivative strains.
|
Findings |
Experimental approach |
Tested strain |
Ref. |
|
Chitinase activity was induced by QS autoinducer C6-HSL
Chitinase activity was blocked in presence of glucose or sucrose
Expression of chitinase enzymes induced by C6-HSL |
Endogenous expression of chitinase in minimal medium supplemented with chitin
Chitinase control under QS autoinducers
Chitinase control under carbon catabolite repression
Enzyme detection after protein resolution on SDS-PAGE |
ATCC 31532
CV026 |
[4] |
|
Chitinase activity was abolished in cviR knockout strain
Chitinase down regulation by QS repressor protein VioS |
Endogenous expression of chitinase in minimal medium supplemented with chitin
Knockout and complementation strains |
ATCC 31532
cviR::Gm knockout strain
vioS::Km knockout strain |
[5] |
|
Chitinase gene (CV_4240) was activated by QS regulator CviR |
Heterologous expression of cviR and PCV_4240::luxCDABE reporter system in E. coli
|
ATCC 12472 |
[21] |
|
Chitinase activity was induced by QS autoinducer C6-HSL
C6-HSL-induced chitinase was blocked by QS antagonist chlorolactone |
Endogenous expression of chitinase in minimal medium supplemented with chitin
QS modulation by autoinducers and antagonists |
CV026 |
[2] |
Specifically regarding chitinase activity in Chromobacterium ssp., we insist that we provided in our response a considerable body of previously published evidence that supports our opinion and endorses the methodological coherence of our experiment, which was criticized on insufficient grounds and without due consideration of the existing literature.
Finally, we have corrected the typographical errors you pointed out.
Thank you for your valuable review and for providing us with the opportunity to have this important scientific discussion
Reviewer 5 Report
Comments and Suggestions for Authors
This review focuses on a response by Dr. Machado's research group to a statement made in the discussion section of a recent paper published by Deryabin et al. (Microorganisms 2025, 13, 1021). Specifically, Dr. Machado's group (Pereira et al. Microorganisms 2025,13, x) rebuts the statements, in this paragraph, "Notably, when comparing the list of QS-controlled genes, chitinase is found among the up-regulated traits in C. violaceum ATCC12472, while it is absent in C. subtsugae ATCC31532. The observation makes questionable the chitinase assay in QS modulation experiments using the ATCC-derived CV026 biosensor strain [55]..... [55] references a prior publication by Dr. Machado's group (Favero et al. RSC Adv. 2023, 26, 18045-18057. Briefly, Dr. Machado's "rebuttal" as expressed in the current manuscript (Pereira et al., Microorganisms 2025, 13. x) is justified for one specific reason, in particular, i.e., experimental design. In the study on quorum sensing (QS), Deryabin et al. (2025) based their findings on data obtained from the C. subtsugae grown in Luria-Bertani medium that was not supplemented with chitin. It is widely known that expression of chitinase genes is dependent on the presence of chitinaceous substrates in culture media. The work by Deryabin et al. (2025) identified a number of genes in C. subtsugae putatively regulated by QS, not on refining thorough experimentation to acquire specific data related to chitinase expression.
In the brief rebuttal by Pereira et al. a number of important works are nicely cited to support their position.
A deeper analysis by the Reviewer revealed that Machado group offers strong evidence that quorum sensing (QS)—a bacterial communication system—is essential for chitinase production in Chromobacterium subtsugae. They showed that synthetic compounds can block QS by interfering with a specific receptor called CviR, which in turn shuts down genes responsible for producing violacein and chitinase. This suggests that QS acts as a switch that turns on these enzymes, and when QS is inhibited, the bacteria lose key traits linked to virulence.
It seems that the study by Deryabin et al. (2025) generalizes QS inhibition as broadly beneficial or harmless across microbial systems, without acknowledging the strain-specific mechanisms Machado's group carefully outlined. Supporting Machado’s view, Kim et al. (https://doi.org/10.1111/mpp.12379) demonstrated that QS is also crucial for chitinase production and antifungal activity in another Chromobacterium strain. When QS-related genes were knocked out, the bacteria lost their biocontrol abilities, which were restored when the genes were reintroduced. This reinforces the idea that QS is necessary for chitinase expression in certain strains. On the other hand, QS doesn’t function the same way in all bacteria. In Vibrio harveyi, QS actually suppresses chitinase production, while in Vibrio cholerae, QS promotes chitin metabolism and helps the bacteria form biofilms and resist predators https://doi.org/10.1038/ismej.2014.265.
So taken together, it seems these contrasting examples show that QS–chitinase relationships vary widely depending on the species and ecological context. Nevertheless, the Reviewer thinks that Machado’s receptor-specific approach aligns with the genetic evidence in Chromobacterium, and their disagreement with Deryabin et al. (2025) statement (above) appears to be is well-founded. On a broader perspective, it highlights the importance of interpreting QS regulation within the precise biological and experimental framework of each bacterial system.
Author Response
Reviewer 5: This review focuses on a response by Dr. Machado's research group to a statement made in the discussion section of a recent paper published by Deryabin et al. (Microorganisms 2025, 13, 1021). Specifically, Dr. Machado's group (Pereira et al. Microorganisms 2025,13, x) rebuts the statements, in this paragraph, "Notably, when comparing the list of QS-controlled genes, chitinase is found among the up-regulated traits in C. violaceum ATCC12472, while it is absent in C. subtsugae ATCC31532. The observation makes questionable the chitinase assay in QS modulation experiments using the ATCC-derived CV026 biosensor strain [55]..... [55] references a prior publication by Dr. Machado's group (Favero et al. RSC Adv. 2023, 26, 18045-18057. Briefly, Dr. Machado's "rebuttal" as expressed in the current manuscript (Pereira et al., Microorganisms 2025, 13. x) is justified for one specific reason, in particular, i.e., experimental design. In the study on quorum sensing (QS), Deryabin et al. (2025) based their findings on data obtained from the C. subtsugae grown in Luria-Bertani medium that was not supplemented with chitin. It is widely known that expression of chitinase genes is dependent on the presence of chitinaceous substrates in culture media. The work by Deryabin et al. (2025) identified a number of genes in C. subtsugae putatively regulated by QS, not on refining thorough experimentation to acquire specific data related to chitinase expression.
In the brief rebuttal by Pereira et al. a number of important works are nicely cited to support their position.
A deeper analysis by the Reviewer revealed that Machado group offers strong evidence that quorum sensing (QS)—a bacterial communication system—is essential for chitinase production in Chromobacterium subtsugae. They showed that synthetic compounds can block QS by interfering with a specific receptor called CviR, which in turn shuts down genes responsible for producing violacein and chitinase. This suggests that QS acts as a switch that turns on these enzymes, and when QS is inhibited, the bacteria lose key traits linked to virulence.
It seems that the study by Deryabin et al. (2025) generalizes QS inhibition as broadly beneficial or harmless across microbial systems, without acknowledging the strain-specific mechanisms Machado's group carefully outlined. Supporting Machado’s view, Kim et al. (https://doi.org/10.1111/mpp.12379) demonstrated that QS is also crucial for chitinase production and antifungal activity in another Chromobacterium strain. When QS-related genes were knocked out, the bacteria lost their biocontrol abilities, which were restored when the genes were reintroduced. This reinforces the idea that QS is necessary for chitinase expression in certain strains. On the other hand, QS doesn’t function the same way in all bacteria. In Vibrio harveyi, QS actually suppresses chitinase production, while in Vibrio cholerae, QS promotes chitin metabolism and helps the bacteria form biofilms and resist predators https://doi.org/10.1038/ismej.2014.265.
So taken together, it seems these contrasting examples show that QS–chitinase relationships vary widely depending on the species and ecological context. Nevertheless, the Reviewer thinks that Machado’s receptor-specific approach aligns with the genetic evidence in Chromobacterium, and their disagreement with Deryabin et al. (2025) statement (above) appears to be is well-founded. On a broader perspective, it highlights the importance of interpreting QS regulation within the precise biological and experimental framework of each bacterial system.
Response to Reviewer 5
Dear Reviewer,
We are grateful for your review of our manuscript. We are pleased to know that our paper has been well-evaluated by our peers. Thank you.
Round 2
Reviewer 4 Report
Comments and Suggestions for Authors
The concerns for this manuscript have been very well taken care of, I would recommend its publication in this journal.